



# Comparison of two closed-path cavity based spectrometers for measuring air-water $CO_2$ and $CH_4$ fluxes by eddy covariance

Mingxi Yang[1], John Prytherch[2], Elena Kozlova[3], Margaret J. Yelland[4], Deepulal Parenkat Mony[5], Thomas G. Bell[1]

[1] Plymouth Marine Laboratory, Prospect Place, Plymouth, UK
[2] Institute for Climate and Atmospheric Science, School of Earth and Environment, University of Leeds, Leeds, UK
[3] College of Life and Environmental Sciences, University of Exeter, North Park Road, Exeter, UK
[4] National Oceanography Centre, European Way, Southampton UK
[5] Inter University Centre for Development of Marine Biotechnology, School of Marine Sciences, Cochin University of Science and Technology, India

*Correspondence to*: Mingxi Yang (miya@pml.ac.uk)

**Abstract.** In recent years several commercialized closed-path cavity based spectroscopic instruments designed for eddy covariance flux measurements of carbon dioxide ($CO_2$), methane ($CH_4$), and water vapor ($H_2O$) have become available. Here we compare the performance of two state-of-the-art models – the Picarro G2311-f and the Los Gatos Research (LGR) FGGA at a coastal site. Both instruments can compute dry mixing ratios of $CO_2$ and $CH_4$ based on the concurrently measured $H_2O$. Additionally, we used a high throughput Nafion dryer to physically remove $H_2O$ from the Picarro air stream. Observed air-sea $CO_2$ and $CH_4$ fluxes from these two analyzers, averaging about 12 mmol m$^{-2}$ d$^{-1}$ and 0.12 mmol m$^{-2}$ d$^{-1}$ respectively, agree within the measurement uncertainties. For the purpose of quantifying dry $CO_2$ and $CH_4$ fluxes, the numerical $H_2O$ corrections appear to be effective and lead to results that are comparable to physical removal of $H_2O$ with a Nafion dryer. We estimate the high frequency attenuation of fluxes in our closed-path setup, which was relatively small (≤10%) for $CO_2$ and $CH_4$ but very large for the much stickier $H_2O$. The Picarro showed significantly lower noise and flux detection limits than the LGR. The hourly flux detection limit for the Picarro was about 2 mmol m$^{-2}$ d$^{-1}$ for $CO_2$ and 0.02 mmol m$^{-2}$ d$^{-1}$ for $CH_4$. For the LGR these detection limits were about 8 mmol m$^{-2}$ d$^{-1}$ and 0.05 mmol m$^{-2}$ d$^{-1}$. Using global maps of monthly-mean air-sea $CO_2$ flux as reference, we estimate that the Picarro and LGR can resolve hourly $CO_2$ fluxes from roughly 40% and 4% of the world's oceans, respectively. Averaging over longer timescales would be required in regions with smaller fluxes. Hourly flux detection limits of $CH_4$ from both instruments are generally higher than the expected emissions from the open ocean, though the signal to noise of this measurement may improve closer to the coast.





## 1 Introduction

Eddy covariance is a direct and non-intrusive method for quantifying the vertical transport of carbon dioxide ($CO_2$) and methane ($CH_4$). This micro-meteorological technique derives fluxes from rapid fluctuations in the atmospheric $CO_2$ and $CH_4$, and thus requires a high frequency (typically 10 Hz) chemical sensor. Over the last couple of decades, open path infrared gas analyzers

(IRGAs), such as the LI7500 (LI-COR Biosciences, Lincoln, Nebraska, USA), have been widely used to measure the atmosphere-biosphere as well as atmosphere-ocean exchange of $CO_2$. A similar instrument (LI7700) is available for measurements of $CH_4$ flux. The advantages of open-path sensors include small size, low power consumption, minimal lag time between chemical and wind sensors, and no high frequency signal attenuation from the inlet tube.

     Miller et al. (2010), Blomquist et al. (2014), and Landwehr et al. (2014) demonstrated that air-sea $CO_2$ fluxes measured

by LI7500s are subject to substantial biases due to variations in water vapor ($H_2O$). When measuring air-sea $CO_2$ fluxes, which are typically on the order of a few mmol m$^{-2}$ d$^{-1}$, bias in the measured $CO_2$ fluxes due to $H_2O$ can be 1 to 2 orders of magnitude greater than the actual $CO_2$ flux signal. This measurement bias is not related to the application of the air density correction (i.e. Webb et al. 1980); rather, it seems to be caused by cross-sensitivities between $CO_2$ and $H_2O$ in the forms of spectral interference and pressure broadening (Kondo et al. 2014). The low flux magnitude of $CO_2$ over the ocean and high humidity exacerbate this

measurement bias. Furthermore, there appears to be little consistency even within the LI-COR family of products, with the LI7500 and the more recent LI7200 showing opposite signs in the $H_2O$ bias (Landwehr et al. 2014). Uncertainties and non-linearity in these cross-sensitivities complicate any mathematical corrections that try to address them (e.g. Prytherch et al. 2010; Edson et al. 2011). Miller et al. (2010) first showed that by converting the open path LI7500 to a closed-path configuration and physically drying the sampled air with a Nafion dryer, the accuracy and precision of the covariance $CO_2$ flux is significantly

improved. The benefit of removing $H_2O$ from the sampled air when measuring air-sea $CO_2$ fluxes with IRGAs is rigorously and quantitatively confirmed by Blomquist et al. (2014) and Landwehr et al. (2014).

     In more recent years, Cavity Ringdown Spectroscopy (CRDS; O'Keefe and Deacon, 1988) as well as Off-Axis Integrated-Cavity Output Spectroscopy (OA-ICOS; O'Keefe et al. 1999) have been developed and commercialized to measure $CO_2$ and $CH_4$. Two of the leading manufacturers are Picarro Inc., Santa Clara, California, USA (CRDS) and Los Gatos Research

(LGR), Mountain View, California, USA (OA-ICOS). These instruments regulate the cavity (i.e. measurement cell) temperature and pressure, obviating the parts of the Webb et al. (1980) correction that are caused by fluctuations in air temperature and pressure. Blomquist et al. (2014) coupled a prototype Picarro CRDS instrument (G1301-f) to a Nafion Dryer and measured $CO_2$ fluxes with an order of magnitude better precision (and free from $H_2O$-bias) compared to measurements with open-path IRGAs (see more details in Section 3.2).





The latest models from Picarro (G2311-f) and LGR (enhanced performance Fast Greenhouse Gas Analyzer, FGGA) measure $CO_2$, $CH_4$, and $H_2O$ simultaneously at 10 Hz. Observed $H_2O$ mixing ratio is used to numerically correct on a point-by-point basis for the effect of humidity on the $CO_2$ and $CH_4$ signals by the instruments' internal software. This in theory enables the derivation of dry $CO_2$ and $CH_4$ fluxes without the need for a physical dryer (e.g. Nafion). The LGR FGGA is a fairly recent

instrument and we are not aware of any published eddy covariance $CO_2$ or $CH_4$ fluxes using this instrument at sea. To the best of our knowledge, measurements of air-water fluxes Picarro and the LGR analyzers have never been compared side-by-side.

Previous tests over terrestrial regions of very high fluxes are briefly reviewed here but do not provide sufficient information with respect to the suitability (or preference) of the Picarro and LGR instruments for air-sea flux measurements. Peltola et al. (2014) compared the performance of eight different instruments for eddy covariance $CH_4$ flux measurements over

grassland, including the Picarro G2311-f and the LGR FGGA. They relied on the numerical $H_2O$ corrections from the Picarro and the LGR instruments for the $CH_4$ flux calculations and did not look at the effect of using a Nafion dryer. The authors found that in this environment with a typical $CH_4$ emission on the order of 2 mmol m$^{-2}$ d$^{-1}$, cumulative measured fluxes from the G2311-f and the FGGA over 14 days agreed within 3% (~0.8 mmol m$^{-2}$). Tuzson et al. (2010) found good agreement between an earlier generation LGR instrument (with numerical $H_2O$ correction) and a quantum cascade laser based absorption

spectrometer (physically dried) when measuring artificially generated $CH_4$ fluxes (> 1 mmol m$^{-2}$ d$^{-1}$) by eddy covariance. In contrast to these areas of high fluxes, estimated $CH_4$ emissions over the open ocean are on the order of 0.01 mmol m$^{-2}$ d$^{-1}$ (e.g. Bange et al. 2006; Forster et al. 2009), placing greater demand on instrument accuracy and precision.

In this work, we compare the performance of the Picarro G2311-f and the LGR enhanced performance FGGA at measuring air-water fluxes of $CO_2$ and $CH_4$ by eddy covariance. Measurements were made from a coastal site, free from

platform motion that might interfere with the flux signal (e.g. on ships or buoys, Prytherch et al., 2015). We compare the precision and flux detection limits of the two analyzers, quantify the effects of $H_2O$ fluctuations on the measured $CO_2$ and $CH_4$ fluxes, and also examine the necessity of drying sample air when using these cavity-based instruments downstream of a long inlet. The gas flow rate required by the Picarro is fairly low (owing to the small cavity size), enabling efficient physical removal of water vapor from the sampled air with a dryer. The LGR has a much larger cavity cell that requires a higher gas flow to

minimize high frequency flux attenuation. The high flow rate results in decreased dryer performance and the LGR was thus configured without a dryer.

## 2 Experimental

### 2.1 Instrumental Setup



The Picarro G2311-f and the LGR FGGA were deployed side-by-side between 25 September and 2 October 2015 at the Penlee Point Atmospheric Observatory (PPAO; Figure 1) on the southwest coast of the United Kingdom. Yang et al. (2016) provided detailed descriptions of this site and reported air-sea $CO_2$ and $CH_4$ flux measurements from the open water wind sector (i.e. southwest) during spring/summer 2014/2015 using the Picarro G2311-f. During the week-long intercomparison described here, wind direction was predominantly from the northeast (Figure A1) over the Plymouth Sound with a wind fetch over water of 5–6 km. Within this wind sector, measured momentum and sensible heat flux suggest that the flux footprint was also over the water only (Appendix, Figures A2 and A3).

Both analyzers sampled from the same ~18 m long PFA inlet tube (3/8'' outer diameter), which led from inside the observatory to a sonic anemometer (Gill Windmaster Pro) on a retractable mast above the rooftop (~18 m above mean sea level). The gas inlet tip was about 30 cm below the sonic anemometer centre volume. A small, stainless steel particle filter (2 µm pore size, Swagelok SS-6F-05) was installed inline to protect the gas sensors from sea salt. An external scroll pump (BOC Edwards XDS-35i) was adjusted to pull ~20 SLPM of air through the inlet tubing. About ~15 SLPM went through the cavity of the LGR analyzer, which has a volume of 408 mL. The LGR cavity pressure was held at ~140 Torr (0.184 atmosphere), resulting in a volumetric flow through the cavity of ~80 LPM. At this rate, the flushing time through the LGR cavity was 0.3 s.

The Picarro analyzer subsampled from the main tubing ahead of the LGR at a flow rate of approximately 5 LPM (at atmospheric pressure). Maintained at a pressure of 153 Torr (0.20 atmosphere), the cavity volume of the Picarro (35 mL) is about 10 times smaller than that of the LGR. The flushing time in the cavity is thus less than 0.1 s and the manufacturer's stated response time is 0.2 s or less. Between 25 September and 1 October, a high throughput dryer (Nafion PD-200T-24M) was installed immediately upstream of the Picarro instrument (i.e. not affecting the LGR). The dryer was setup in the reflux configuration, utilizing the reduced pressure of the Picarro exhaust air to dry the sample air. Using the LGR as reference, the dryer eliminated ~80% of $H_2O$ (and ~95% of the variability in $H_2O$) in the sample stream of the Picarro instrument. We will refer to this sample period as Picarro (dry) vs LGR (wet). Between 1 and 2 October, the Nafion dryer was removed in order to briefly examine the humidity dependence in the Picarro analyzer. We will refer to this period as Picarro (wet) vs LGR (wet).

**2.2 Numerical Corrections for Water Vapor**

Both analyzers report ambient mixing ratios of $CO_2$ and $CH_4$ ($C_{CO2}$ and $C_{CH4}$) in parts per million (ppm). They compute the 'dry mixing ratios' ($C_{CO2\_d}$ and $C_{CH4\_d}$, in ppm) based on the measured volume fraction of water vapor ($C_{H2O}$, in %). Water vapor affects $CO_2$ and $CH_4$ measurements in cavity-based instruments in at least two ways: volumetric dilution and spectroscopic line broadening. Rella (2010) proposed the following corrections to account for both effects together:

$$C_{CO2} = C_{CO2\_d} (1 + a\, C_{H2O} + b\, C_{H2O}^2) \tag{1}$$



$$C_{CH4} = C_{CH4\_d} \, (1 + c \, C_{H2O} + d \, C_{H2O}^2) \qquad\qquad (2)$$

For the Picarro G-2311f analyzer, Chen et al. (2010) found a = -0.01200, b = -2.674 x $10^{-4}$, c = -0.00982, d = -2.393 x $10^{-4}$. When air is dried with a Nafion dryer, the reliance of the Picarro on these numerical corrections is reduced by about an order of magnitude. Different coefficients have been estimated for the LGR FGGA (Hiller et al. 2012). However, comparisons

of $C_{CO2}$ vs $C_{CO2\_d}$ as well as $C_{CH4}$ vs $C_{CH4\_d}$ in our observations clearly show that only a dilution correction has been applied internally by this FGGA. In the case of dilution only, a and c = -0.01, while b and d = 0. $C_{H2O}$ is on the order of 1% in this marine environment. Accounting for volumetric dilution thus increases the mean $CO_2$ and $CH_4$ mixing ratios by about 1%, while the correction for line broadening is more than an order of magnitude smaller than the dilution correction. As a result, we expect biases in $CO_2$ and $CH_4$ fluxes computed from Equations 1 and 2 only when the fluctuations in $H_2O$ are large and are correlated

with fluctuations in dry $CO_2$ and $CH_4$ mixing ratios (i.e. due to any residual cross-sensitivity with $H_2O$).

The slope in mixing ratios (intercept, $r^2$) during Picarro (dry) vs. LGR (wet) was 1.014 (-15.5 ppm, 0.99) for $C_{CO2\_d}$ and 0.976 (0.0113 ppm, 0.99) for $C_{CH4\_d}$ (Picarro sampling behind a dryer). The calibrations of these two instruments were also crudely checked with a concentrated calibration gas mixture (8000 ppm $CO_2$ and 40 ppm $CH_4$, BOC gas). $C_{CO2\_d}$ and $C_{CH4\_d}$ from both instruments were within the certified uncertainty of the gas standards (5%) so no calibration curve was applied. For

the Picarro, the use of the Nafion dryer reduced the observed $CO_2$ mixing ratio by ~0.3% and reduced the $CH_4$ mixing ratio by ~0.06% when measuring the concentrated calibration gas mixture. This suggests that small amounts of $CO_2$ and $CH_4$ permeate through the Nafion dryer, qualitatively consistent with results from Welp et al. (2013).

**2.3 Flux Processing**

Fluxes of $CO_2$, $CH_4$, and $H_2O$ are computed using the Picarro and LGR data along with the streamline corrected vertical wind velocity ($w$) from the Windmaster Pro sonic anemometer (see Yang et al. 2016 for further details about flux processing). Lag correlations between $C_{CO2\_d}$, $C_{CH4\_d}$ from the Picarro and LGR with $w$ generally showed a consistent maximum covariance lag time of about 3 s. The maximum covariance lag time between $H_2O$ and $w$ (without a dryer) tended to be much longer and more variable (~20 s) due to severe attenuation of the high frequency water vapor signal in the inlet tube (see Section 3.3).

Covariance from both instruments are computed in the same, non-overlapping 10-minute intervals between lag-shifted gas mixing ratios and $w$ (e.g. $\overline{w'C_{CO2\_d}'}$, $\overline{w'C_{CH4\_d}'}$), which are multiplied by the ambient air density to yield mass concentration fluxes. Here the primes indicate fluctuations from the means while the overbar denotes temporal averaging. The resultant 10-minute fluxes are further averaged to hourly intervals in order to reduce random noise. The relatively short averaging time of 10 minutes is chosen here to more easily satisfy the stationarity flux criteria in this dynamic environment. As





demonstrated by Yang et al. (2016), low frequency flux not captured by the 10-minute averaging window is usually small at this

site (within a few percent) and in any case is the same for both analyzers.

      The LGR showed a positive offset in $C_{CO2\_d}$ of ~15 ppm relative to the Picarro for reasons currently unknown. When

measuring high purity nitrogen, the LGR $CO_2$ signal was also around 15 ppm initially and decayed towards zero after more than

an hour. Since the covariance flux is computed from deviations from linearly-detrended mean in 10-minute intervals, the

positive bias in the LGR $CO_2$ mixing ratio should not significantly affect our flux comparison.

## 3 Results and Discussions

      Time series of $CO_2$ and $CH_4$ flux measurements during the Picarro (dry) vs LGR (wet) period are shown in Figure 2. For the

LGR, both ambient (e.g. from $C_{CO2}$) and numerically dry (e.g. from $C_{CO2\_d}$) fluxes are shown. Considering all wind directions

(137 flux hours), there is good agreement between the two instruments for dry $CO_2$ flux (slope = 0.99) and dry $CH_4$ flux (slope =

1.00). The mean (± standard error) dry $CO_2$ and $CH_4$ fluxes from the Picarro were 27.9 ± 6.4 and 0.167 ± 0.021 mmol m$^{-2}$ d$^{-1}$,

respectively. These values agree within uncertainties with dry $CO_2$ and $CH_4$ fluxes from the LGR, which were 28.3 ± 6.7 and

0.176 ± 0.022 mmol m$^{-2}$ d$^{-1}$, respectively. Likewise, during the Picarro (wet) vs LGR (wet) period, there was no statistically

significant difference between the dry $CO_2$ and $CH_4$ fluxes from the Picarro and the LGR.

      We examine the impact of the numerical $H_2O$ correction on the $CO_2$ and $CH_4$ fluxes within a given instrument. This

'intra-instrument' difference in flux, computed from numerically dry and ambient mixing ratios (i.e. $\overline{w'C_{CH4\_d}'} - \overline{w'C_{CH4}'}$),

is plotted against the apparent $H_2O$ flux (i.e. $\overline{w'C_{H2O}'}$ at the optimal lag time of $CH_4$ / $CO_2$) for both the LGR and Picarro in

Figure 3. For the Picarro, these data are taken from the Picarro (wet) vs LGR (wet) period. The slopes between the 'intra-

instrument' flux differences and $H_2O$ fluxes are steeper for the Picarro than for the LGR. This is consistent with the fact that the

Picarro G2311-f applies both a dilution and a line broadening correction (Equations 1 and 2), while the LGR FGGA applies only

the dilution correction. The slopes in Figure 3 are essentially equal to the slopes expected between the gas mixing ratios as a

result of the numerical $H_2O$ corrections. At typical background levels of these greenhouse gases, the expected slope of [$C_{CO2\_d}$ –

$C_{CO2}$] vs. $H_2O$ mixing ratio is ~0.004 for the LGR and ~0.006 for Picarro; the analogous slope in the case of $CH_4$ is ~1.9e$^{-6}$ for

the LGR and 2.5e$^{-6}$ for the Picarro. Thus it appears that cross-sensitivities between $CO_2$ and $CH_4$ fluxes vs $H_2O$ fluxes are well

accounted for by Equations 1 and 2. There is no additional non-linearity that is detectable in the dry $CO_2$ and $CH_4$ fluxes due to

the presence of $H_2O$ in our data.

### 3.1 Comparisons of Air-Water Fluxes of $CO_2$ and $CH_4$





As indicated in Figure 2 by the gray shading, air-water $CO_2$ and $CH_4$ fluxes tend to be much lower than terrestrial fluxes. To further evaluate the performance of these analyzers at air-sea flux measurements, we limit our comparison to the sector when the wind was coming from over water only. Within the northeast quadrant, we find that the fluxes of momentum and sensible heat are reasonably consistent with air-water transfer when the wind directions were between 45 and 80º (Appendix, Figures A2 and

A3). We compare fluxes of $CO_2$ and $CH_4$ from the Picarro and LGR instruments within this wind sector in Figure 4 (73 flux hours). Here $CO_2$ and $CH_4$ fluxes have further been filtered for staionarity following Yang et al. (2016). This removed occasional periods when the atmospheric $CO_2$ and $CH_4$ mixing ratios were highly variable (e.g. due to large horizontal transport or emissions from passing ships in the Plymouth Sound).

For both gases, fluxes from the two instruments scatter around the 1:1 line and differences in the mean fluxes between
the two instruments are within the measurement uncertainties. During the Picarro (dry) vs LGR (wet) period, the mean (±standard error) dry $CO_2$ flux was $10.7 \pm 1.8$ mmol $m^{-2}$ $d^{-1}$ from the Picarro and $12.5 \pm 2.9$ mmol $m^{-2}$ $d^{-1}$ from the LGR. The dry $CH_4$ flux was $0.110 \pm 0.009$ mmol $m^{-2}$ $d^{-1}$ from the Picarro and $0.123 \pm 0.011$ mmol $m^{-2}$ $d^{-1}$ from the LGR. The root mean squared (RMS) error between the physically dry Picarro flux and the numerically dry LGR flux (hourly average) is 16.1 mmol $m^{-2}$ $d^{-1}$ for $CO_2$ and 0.063 mmol $m^{-2}$ $d^{-1}$ for $CH_4$. The RMS errors increase by only ~1% when comparing the ambient LGR
fluxes to the dry Picarro fluxes. Since the gas fluxes were computed using the same wind data and the two analyzers were sampling the same gas stream, differences between them are primarily caused by noise in the instruments, rather than by the presence of water vapor.

The effects of $H_2O$ on the LGR fluxes were detectable but relatively small compared to the magnitude of the fluxes. This is illustrated in Figure 5, where the differences in $CH_4$ fluxes between the LGR (numerically dry as well as ambient) and
Picarro (physically dry) are plotted against the bulk air-water latent heat flux predicted from the Coupled Ocean-Atmosphere Response Experiment (COARE) model (Fairall et al. 2003). Discrepancies between the two sets of the LGR $CH_4$ fluxes are most apparent (up to 0.04 mmol $m^{-2}$ $d^{-1}$ in either direction) under conditions of high latent heat flux (i.e. highest fluctuations in $H_2O$ mixing ratio). These results are similar to the findings from Tuzson et al. (2010), who showed that with an earlier generation LGR instrument, not accounting for water vapor (either physically through drying or numerically) results in a bias in the eddy
covariance $CH_4$ flux. In our setup, additional application of a spectral line broadening correction has a negligible impact on the LGR $CO_2$ and $CH_4$ fluxes.

Figure 6 shows the mean cospectra of $CO_2$, $CH_4$ (Picarro physically dry, LGR ambient, LGR numerically dry), as well as momentum when winds were from the air-water wind sector. Each cospectrum is normalized by the respective mean flux. The gas cospectra from the two instruments have similar spectral shape to the momentum cospectrum. Numerical drying of the
LGR only has a visible impact on the cospectra at low frequencies, which is likely due to the severe dampening of the $H_2O$ flux





in the main inlet tubing (Section 3.3). The LGR shows much more noise at frequencies above ~0.1 Hz, which is partly caused by the less than optimal ringdown time of the LGR analyzer during our testing period (~8 µs instead of nominal value of ~12 µs). Dust and/or sea salt might have entered the LGR cavity during instrument installation, thereby reducing the ringdown time and instrument sensitivity.

### 3.2 Instrument Noise and Eddy Covariance Flux Detection Limit

Scatter in the hourly LGR flux was about 20 and 50% higher than in the Picarro for $CH_4$ and $CO_2$, respectively. Variance spectra of $C_{CO2\_d}$ and $C_{CH4\_d}$ from the Picarro and LGR show much higher noise in the latter instrument (Figure 7). These were averaged from the Picarro (dry) vs LGR (wet) period for the air-water wind sector. Excluding the small spikes near 3 Hz (likely

instrument artifacts), the mean $CO_2$ variance above 1 Hz (i.e. band-limited noise) was 0.0023 $ppm^2$ $Hz^{-1}$ for the Picarro and 0.3 $ppm^2$ $Hz^{-1}$ for the LGR. In the case of $CH_4$, mean variance above 1 Hz was about 0.23 $ppb^2$ $Hz^{-1}$ for the Picarro and 5 $ppb^2$ $Hz^{-1}$ for the LGR. Interestingly, at low frequencies the LGR variance was also greater than the Picarro (by ~2 times for $CO_2$ and ~60% for $CH_4$). This may be partly because the LGR does not regulate the cavity temperature and pressure (1 σ of about 0.005 °C and 0.16 mbar within an hour) as precisely as the Picarro (1 σ of about 0.0007 °C and 0.06 mbar). Low frequency

fluctuations in $H_2O$, if not fully accounted for in the computations of the dry mixing ratios, could also cause some apparent variance in $CO_2$ and $CH_4$ for the LGR.

Uncertainty and so detection limit in eddy covariance gas fluxes depend upon variance in both the vertical wind velocity as well as in the gas mixing ratio. The latter is the sum of the ambient variance of the gas (estimated as the second point of the autocovariance of the mixing ratio; see Blomquist et al. 2010) and instrumental noise (estimated as the difference between the

first and second points of the autocovariance). Both the Picarro and the LGR instruments are able to resolve rapid ambient fluctuations in atmospheric $CO_2$ and $CH_4$. For example, ambient variance in $C_{CH4\_d}$ (i.e. without instrument noise) from the two instruments agree almost exactly (slope = 1.00; $r^2$ = 0.96).

Based on earlier measurements at PPAO, Yang et al. (2016) estimated a $CH_4$ flux detection limit of 0.02 mmole $m^{-2}$ $d^{-1}$ (hourly average) for the Picarro G2311-f at a wind speed of 10 m $s^{-1}$. This was quantified using two different methods that

yielded comparable results: theoretically based on instrument noise and ambient variance (Blomquist et al. 2014), and empirically based on scatter in $\overline{w'C_{CH4}'}$ at an implausible lag time (i.e. 300 s). Applying the same methods to $CO_2$ here results in a flux detection limit between 2.2 mmole $m^{-2}$ $d^{-1}$ (theoretical) and 4 mmole $m^{-2}$ $d^{-1}$ (empirical) for the G2311-f. At a wind speed of 10 m $s^{-1}$ and in a neutral atmosphere, Blomquist et al. (2014) computed a $CO_2$ flux detection limit of 1.3 mmole $m^{-2}$ $d^{-1}$ (hourly average) for a prototype Picarro analyzer G1301-f. The instrument used by Blomquist et al. (2014) only measures $CO_2$

and thus has a lower noise level of 700 $ppb^2$ $Hz^{-1}$ (see their Figure 6) than the Picarro we deployed at PPAO.





Following Blomquist et al. (2014), we theoretically estimate the flux detection limits for the LGR FGGA using the band-limited noise shown in Figure 7 and ambient variance from periods of very low variability (100 and 0.04 $ppb^2$ for $CO_2$ and $CH_4$, respectively). At a wind speed of 10 m $s^{-1}$ and in a neutral atmosphere, we estimate a flux detection limit of 13.5 mmole $m^{-2}$ $d^{-1}$ for $CO_2$ and 0.060 mmole $m^{-2}$ $d^{-1}$ for $CH_4$ (hourly average). We also examined an earlier measurement period when the

LGR had the optimal sensitivity (ringdown time of 12 μs), with band-limited noise of 0.11 $ppm^2$ $Hz^{-1}$ for $CO_2$ and 3.7 $ppb^2$ $Hz^{-1}$ for $CH_4$. Coupling these noise levels to the ambient variability above, at the same environmental conditions we estimate a best case hourly flux detection limit for the LGR of 8.2 mmole $m^{-2}$ $d^{-1}$ for $CO_2$ and 0.053 mmole $m^{-2}$ $d^{-1}$ for $CH_4$ (see Table 1 for summary). In the case of $CO_2$, the LGR flux detections limits are of the same magnitude as those from the LI-COR LI7200 with a dryer (as estimated by Blomquist et al. 2014).

It is useful to put these flux detection limits into the context of regions of the ocean and times of year that are favorable for future process-level studies with direct flux measurements (e.g. research cruises that aim to improve our understanding of the gas transfer velocity). For illustrative purposes, we compare the $CO_2$ flux detection limits with the estimated global air-sea $CO_2$ flux maps from the National Oceanographic and Atmospheric Administration's Pacific Marine Environmental Laboratory (http://www.pmel.noaa.gov/co2/story/Surface+CO2+Flux+maps; Feely et al. 2006; Sabine et al. 2007, 2009; Park et al. 2010).

Four monthly maps from 2015 to 2016 (4° resolution) are chosen to capture seasonal variations: March, June, September, and December. Globally, the absolute value of the estimated monthly-mean $CO_2$ flux exceeds 2 mmole $m^{-2}$ $d^{-1}$ (hourly flux detection limit of the Picarro) in roughly 39–44% of the oceans (highest in March), and exceeds 8 mmole $m^{-2}$ $d^{-1}$ (detection limit of the LGR) in 2–6% of the oceans (highest in December). The Northern Atlantic and Pacific, Southeastern Pacific, and the Southern Ocean are not surprisingly amongst regions favorable for eddy covariance $CO_2$ flux measurements. When estimated fluxes are

comparable or below the hourly flux detection limit, further averaging of the eddy covariance measurements (temporally or in bins of wind speed, etc) will likely be necessary to extract statistically meaningful results (i.e. random uncertainty in flux decreases with increasing number of independent hourly measurements, N, by the relationship of $N^{-0.5}$). For example, if the hourly fluxes are binned to 6-hour intervals the percentage of exceedance as described above may improve to 65% and 24% for the Picarro and LGR, respectively (averaged across all seasons).

The predicted open ocean emission of $CH_4$ is on the order of 0.01 mmol $m^{-2}$ $d^{-1}$ (Forster et al. 2009). Thus further binning of hourly eddy covariance measurements (e.g. to 6-hour or daily intervals) would likely be needed to better resolve $CH_4$ fluxes. The signal to noise ratio in the eddy covariance $CH_4$ flux measurement should be improved near the coast (e.g. Bange et al. 2006) and in the Arctic (e.g. Shakhova et al. 2010), where surface saturations and hence emissions of $CH_4$ are likely greater. Combining data from a floating chamber with a turbulent diffusivity model, Kitidis et al. (2007) estimated a $CH_4$ emission of

0.06 to 0.17 mmole $m^{-2}$ $d^{-1}$ in a large coastal embayment. $CH_4$ flux of a similar magnitude has been measured here at PPAO (this





paper and Yang et al. 2016). Aircraft observations suggest $CH_4$ emissions over 0.1 mmole $m^{-2}$ $d^{-1}$ from the partially ice-covered

Arctic (Kort et al. (2012), which are above the flux detection limits of both analyzers.

We did not evaluate the effect of motion on these analyzers. During the High Wind Gas Exchange Study (HiWinGS)

cruise in 2013 (Yang et al. 2014b), a Picarro G1301-f was deployed with a Nafion dryer along with a second generation model

from Picarro (G2301-f) that was not sampling after a dryer. In moderate seas, the two instruments yielded similar $CO_2$ fluxes,

implying that the numerical $H_2O$ correction in G2301-f is reasonable. However, the different sensitivities of these two models to

the ship's motion in high seas complicated these flux comparisons (B. Blomquist, personal communications, 2016). Side-by-side

shipboard deployments of the Picarro and LGR are required to provide a more conclusive verdict on the optimal flux analyzer for

air-sea flux measurements.

**3.3 High Frequency Flux Loss**

Based on measurements of dimethylsulfide (DMS) flux, Blomquist et al. (2010) and Yang et al. (2011) reported high frequency

flux attenuations to be on the order of 5% using a long inlet and the same type of Nafion dryer as used here for the Picarro. $CO_2$

and $CH_4$ are less polar (i.e. less 'sticky') gases than DMS so we expect their flux losses in the dryer to be comparable or less.

$CO_2$ and $CH_4$ flux attenuation by the tubing itself should also be insignificant given the turbulent flow used in our study

(Lenschow and Raupach, 1991). Below we quantify the high frequency attenuations in the $CO_2$ and $CH_4$ fluxes from the Picarro

(with a dryer) and the LGR (without a dryer).

The effect of the Nafion dryer on the Picarro instrument response is clearly illustrated by making rapid, step-wise

reductions in the $CO_2$ and $CH_4$ gas standard flow rates (here controlled by a digital mass flow controller, EL-FLOW, Bronkhorst)

during calibration with and without the dryer (Figure 8). For ease of comparison, we normalize the dry $CO_2$ and $CH_4$ such that

the signals decrease from one to zero. Without the dryer, the observed response time (defined as the time needed for the signal

to fall to 1/e of the initial value) was about 0.2 s. This is 0.1 s longer than the expected response time given the flow rate, likely

in part due to the finite time response of the mass flow controller (settling time of 0.1–0.2 s). With the dryer, the signal drop-off

was noticeably slower and the observed response time increased to about 0.5 s. The response time for $CO_2$ appears to be very

slightly longer than for $CH_4$ in the presence of the dryer, as might be expected from the greater permeation (i.e. breakthrough) of

$CO_2$ through the Nafion membranes compared to $CH_4$. Knowing the instrument response time allows us to estimate the high

frequency flux loss using a filter function (e.g. Bariteau et al. 2010; Yang et al. 2014a; Blomquist et al. 2014). At a response

time of 0.3 s with the dryer (accounting for the settling time of the mass flow controller), the predicted flux loss in the Picarro

averaged over the Picarro (dry) vs LGR (wet) period (mean wind speed of 10 m $s^{-1}$) is 10%.



Compared to the momentum cospectrum, the Picarro $CO_2$ and $CH_4$ cospectra are noticeably attenuated above the frequency of about 1 Hz (Figure 6). High frequency loss in the gas fluxes is often estimated by similarity scaling with respect to another variable. Here we convert the cospectra of gases and momentum to ogive (Oncley, 1989), or the cumulative sum of the cospectrum from low to high frequency. Following the method outlined by Spirig et al. (2005), from the ogive we estimate that

the mean loss of flux is about 6% for $CO_2$ and 4% for $CH_4$, somewhat less than the filter function estimates above.

For the LGR, most of the flux loss in our setup should occur within the large volume of the cavity cell. Judging from the cospectra, the $CO_2$ and $CH_4$ flux attenuation in the LGR is comparable to the Picarro with a Nafion dryer. We note that the flow rate through the LGR can be increased substantially (by approximately a factor of seven) relative to the setup here by adjusting the external scroll pump and using an inlet tube with a larger inner diameter. A faster airflow through the LGR cavity

(e.g. flushing time of 0.1 s) would reduce the estimated flux loss in the LGR analyzer to only a few percent.

Unlike $CO_2$ and $CH_4$, fluctuations in water vapor are severely attenuated in the main inlet tube (e.g. Ibrom et al. 2007). This is because the polar $H_2O$ is a much 'stickier' gas than $CO_2$ and $CH_4$ and tends to absorb onto the wall of the tubing. Figure 9 shows the mean cospectrum of LGR $H_2O$, which was computed at the maximum covariance lag time of $CO_2$ / $CH_4$ (3 s) as well as at the optimal lag time for $H_2O$ (~20 s). $H_2O$ cospectrum computed at a lag time of 3 s is positive at the lowest frequencies

and negative between about 0.01 and 0.1 Hz (resulting in mean apparent $H_2O$ flux near zero). In reality, the calculated bulk latent heat flux was always positive during this period, averaging 120 W m$^{-2}$ (range of 60–230 W m$^{-2}$).

Computed at the optimal lag time of ~20 s, $H_2O$ cospectrum from the LGR is positive at frequencies below ~0.1 Hz. Above 0.1 Hz the measured $H_2O$ flux is essentially zero (i.e. flux completely attenuated). Comparison with the bulk latent heat flux suggests that only ~10% of the $H_2O$ flux is still detectable after going through ~18 m of inlet tubing. $H_2O$ flux from the

Picarro without the Nafion dryer yielded similar results, implying that it was the tubing rather than the large cavity of the LGR that attenuated most of the $H_2O$ flux. In our setup, $H_2O$ flux attenuation might be particularly severe because the tubing we used had been constantly exposed to marine air for 1.5 years, resulting in accumulation of hygroscopic sea salt particles on the tubing wall. In a case of a shorter, cleaner inlet tubing, dampening of $H_2O$ fluctuations in the tubing should be somewhat reduced and the impact of $H_2O$ on $CO_2$ and $CH_4$ fluxes may be greater. Without a physical drier, this would increase the reliance on the

numerical $H_2O$ correction for accurate determination of $CO_2$ and $CH_4$ fluxes.

## 4 Conclusions

In this paper we compared the performance of the Picarro G2311-f (with and without a Nafion dryer) and the Los Gatos Research FGGA (without a dryer) at eddy covariance measurements of air-water $CO_2$ and $CH_4$ fluxes. Within measurement

uncertainties, dry $CO_2$ and $CH_4$ fluxes from these two analyzers agree. For winds over the Plymouth Sound, air-water $CO_2$ and



$CH_4$ fluxes averaged about 12 mmol m$^{-2}$ d$^{-1}$ and 0.12 mmol m$^{-2}$ d$^{-1}$, respectively. The numerical corrections of $CO_2$ and $CH_4$ mixing ratios based on concurrently measured $H_2O$ appeared to be effective for both instruments. Numerically dry $CO_2$ and $CH_4$ fluxes from the LGR compare well with results from the Picarro after physical removal of $H_2O$ using a Nafion dryer. Coupling the LGR to a dryer, which we did not test, may have slightly improved the instrument precision. The addition of a dryer would

also reduce the reliance on the instrumental numerical correction for $H_2O$, but could increase flux attenuation and potentially cause a small bias in the mean $CO_2$ and $CH_4$ mixing ratios. Flux attenuation in the Picarro (occurring primarily within the dryer) was within 10% for both $CO_2$ and $CH_4$. Similar flux losses were observed for the LGR without a dryer in our setup (occurring primarily within the large cavity), which could be reduced by increasing the airflow through the instrument. In contrast to $CO_2$ and $CH_4$, $H_2O$ fluctuations were severely dampened in our inlet tubing, resulting in ~90% loss of the latent heat flux. Compared

to the LGR, the Picarro demonstrated significantly lower noise levels and so flux detection limits (Table 1). These estimates help to advise on the choices of location and timing for future process-level studies on air-sea $CO_2$ and $CH_4$ transfer.

**Appendix: Selection of Wind Sector for Air-Water Transfer**

The PPAO site is exposed to marine air over a wide wind sector from about 30 to 250º (Figure 1). Wind was mostly from the

northeast during this period of instrument intercomparison (Figure A1). Within this quadrant, we find that the fluxes of momentum (Figure A2) as well as sensible heat (Figure A3) are reasonably consistent with air-water transfer when the wind directions were from 45–80º. As shown by the inset in Figure 1, the distance between PPAO and the water's edge is about 20–30 m towards the northeast. North of a wind direction of 45º, the flux footprint begins to overlap with land (Mount Edgcumbe Country Park). Immediately south of 80º, the longer foreshore in front of PPAO likely affects the flux footprint and

increases the effect of airflow distortion on the measured wind speed. The 10-m neutral drag coefficient is computed as

$$C_{D10N} = (u_* / U_{10N})^2).$$ Here $u_*$ is the friction velocity measured by eddy covariance and the 10-m neutral wind speed $U_{10N}$ is determined using Businger-Dyer relationships (Businger, 1988) from the wind speed and air temperature at PPAO, tidal-dependent sampling height, and sea surface temperature (SST) from the L4 mooring station (~6 km south of PPAO). Within 45–80º, $C_{D10N}$ increases with wind speed, as expected for air-water transfer. Values of $C_{D10N}$ are however about 15% lower than

parameterizations for open water in the mean (e.g. COARE model version 3.5, Edson et al. 2013; Smith 1980), possibly due to flow distortion in mean wind speed or a residual bias in the Windmaster Pro wind measurement (see Yang et al. 2016 for discussion).

Sensible heat flux was computed from the sonic temperature and further corrected for the bulk latent heat contribution. It shows a surprisingly good agreement (slope = 0.98 and r$^2$ = 0.87) with sensible heat flux estimated with the COARE model

(Fairall et a. 2003) using air temperature and wind speed at PPAO as well as SST from L4. This close agreement despite the





relatively long distance between the flux footprint and the mooring station was possibly due to fair weather in September 2015 and minimal riverine influence in the flux footprint (i.e. negligible spatial gradient in SST). Sensible heat flux was on average positive at night and negative during the day, primarily due to diel variability in air temperature. In spite of this variation, the atmosphere was near neutral during this measurement period as a result of the relatively strong winds (mean of 10 m s$^{-1}$). The

predicted atmospheric stability parameter $z/L$ varied from about -0.20 at night to 0.05 during the day. Thus we do not expect the flux footprint to extend beyond the 5–6 km fetch of water to the opposite side of the Plymouth Sound (as could happen during periods of extreme stability). Note that a flux footprint extending over land would cause momentum and sensible heat fluxes to grossly deviate from the predicted bulk air-sea fluxes, which was not observed.

**Acknowledgment**

This work is a contribution to the ACSIS (The North Atlantic Climate System Integrated Study) and ORCHESTRA **(**Ocean Regulation of Climate through Heat and Carbon Sequestration and Transports**)** project funded by the Natural Environment Research Council, UK. Trinity House (http://www.trinityhouse.co.uk/) owns the Penlee site and has kindly agreed to rent the building to PML so that instrumentation can be protected from the elements. We are able to access the site thanks to the

cooperation of Mount Edgcumbe Estate (http://www.mountedgcumbe.gov.uk/). We thank A. J. Watson (University of Exeter), P. D. Nightingale and T. J. Smyth (PML) for support, as well as B. W. Blomquist (US National Oceanographic and Atmospheric Administration) for helpful comments and suggestions. The participation of Deepulal P. M. was funded by the Partnership for Observation of the Global Oceans (POGO) program.

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



Table 1. Performance of the Picarro G2311-f and LGR FGGA at measuring $CO_2$ and $CH_4$ fluxes by eddy covariance. Precision refers to observed noise at 10 Hz. Band-limited noise is averaged between 1 and 5 Hz from the mean variance spectra. Flux detection limit is estimated at a wind speed of 10 m s$^{-1}$ in a neutral atmosphere (hourly average). The FGGA band-limited noise and detection limit correspond to when the ringdown time was optimal and the instrument sensitivity at its highest.

| $CO_2$ | Precision (10 Hz) [ppm] | Band-limited Noise [ppm$^2$ Hz$^{-1}$] | Flux Detection Limit [mmol m$^{-2}$ d$^{-1}$] |
|---|---|---|---|
| Picarro G2311-f | 0.15 | 0.0023 | 2 |
| LGR FGGA | 1.4 | 0.11 | 8 |

| $CH_4$ | Precision (10 Hz) [ppb] | Band-limited Noise [ppb$^2$ Hz$^{-1}$] | Flux Detection Limit [mmol m$^{-2}$ d$^{-1}$] |
|---|---|---|---|
| Picarro G2311-f | 1.1 | 0.23 | 0.02 |
| LGR FGGA | 5.5 | 3.7 | 0.05 |

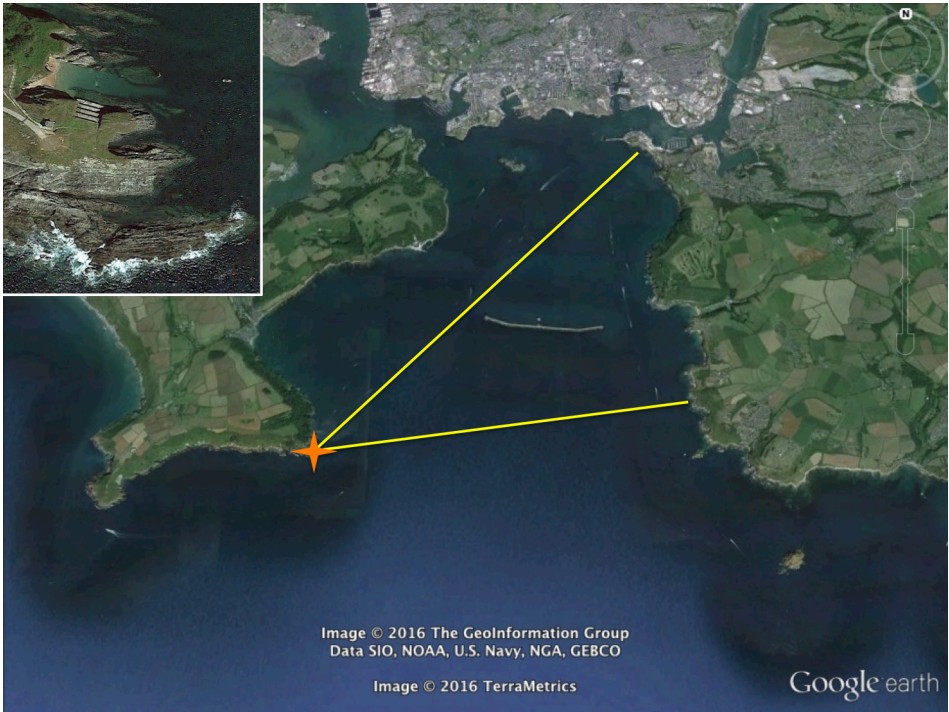

Figure 1. Location of Penlee Point Atmospheric Observatory (orange 4-point star). The yellow lines mark the air-water wind
10    sector of 45 to 80°. The inset shows the area around the observatory (small hut in the left-centre of the image).





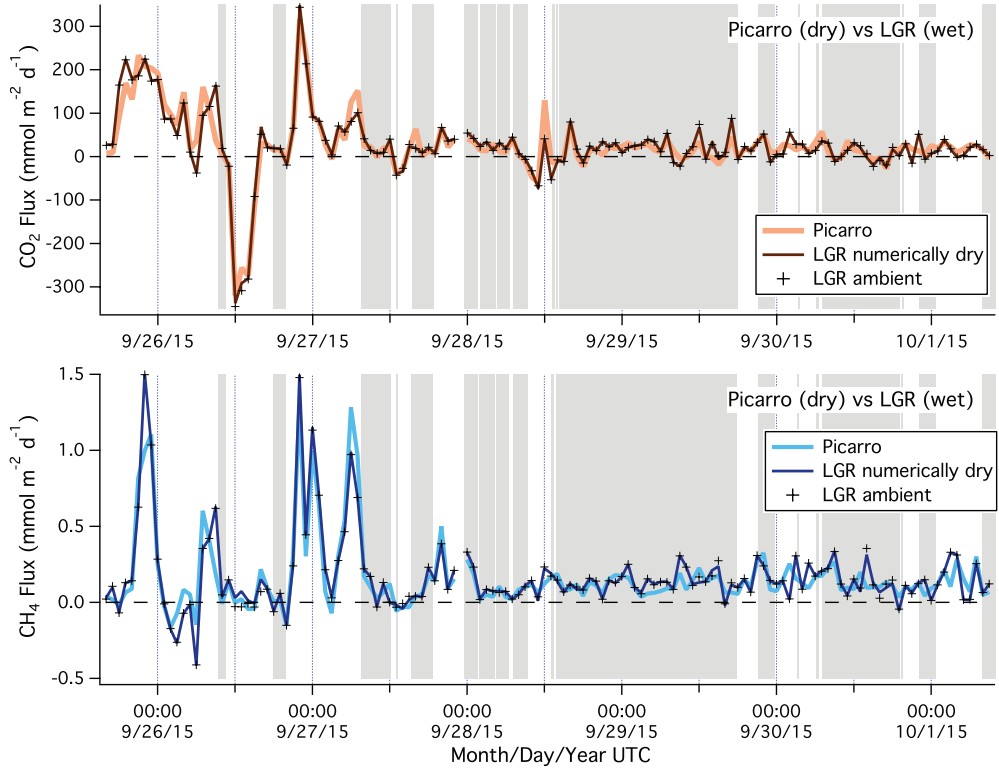

Figure 2. Time series of $CO_2$ (top) and $CH_4$ (bottom) fluxes. Picarro sampled after a dryer, while LGR sampled ambient (moist) air. 'LGR ambient' indicates flux prior to the numerical $H_2O$ correction. The shaded region indicates the wind sector between 45 and 80°.

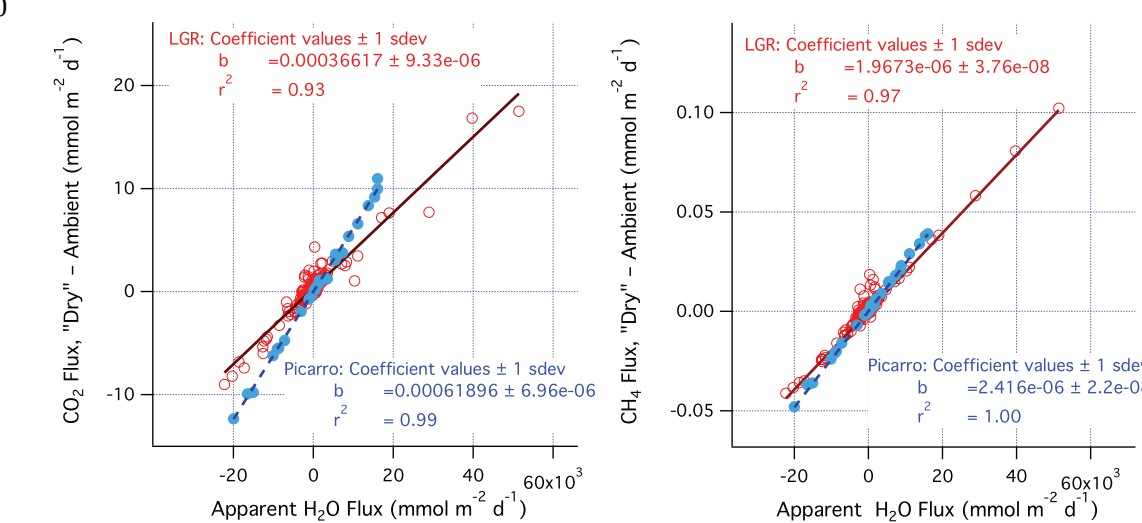

Figure 3. 'Intra-instrument' difference in $CO_2$ (left) / $CH_4$ (right) fluxes computed from numerically dry and ambient $CO_2$ / $CH_4$ mixing ratios from the same instrument vs. apparent $H_2O$ flux (latter computed at the lag time of $CO_2$ / $CH_4$). Both the LGR (red) and Picarro (blue) were directly sampling ambient air. Slopes (b) and $r^2$ values from the linear regressions are also shown.





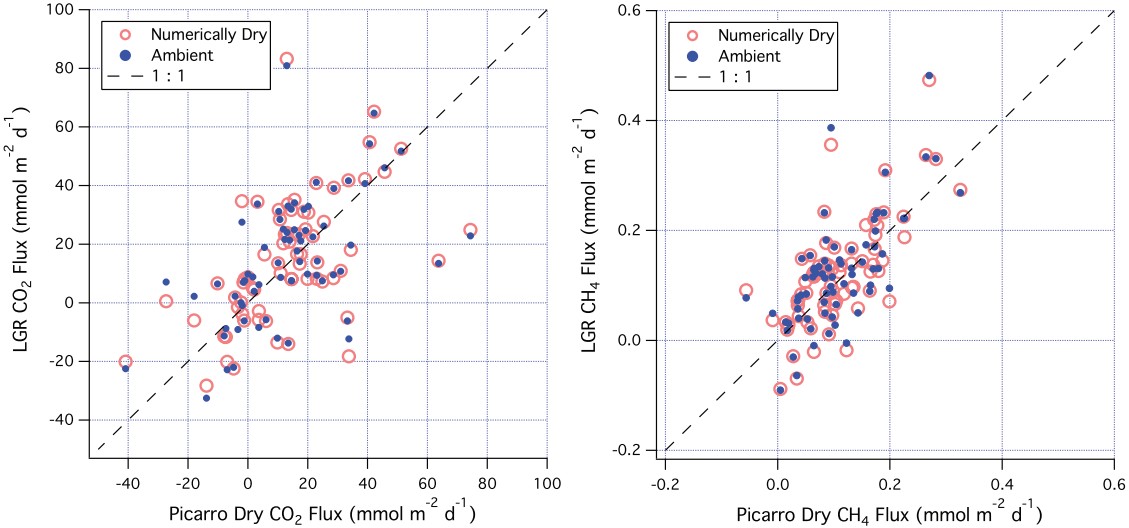

Figure 4. LGR (numerically dry and ambient) vs. Picarro (physically dry) $CO_2$ (left) / $CH_4$ (right) fluxes for the air-water wind
sector of 45 to 80°. For $CO_2$, RMS error between the two instruments is 16.1 mmol m$^{-2}$ d$^{-1}$ using numerically dry LGR data and
16.3 mmol m$^{-2}$ d$^{-1}$ using ambient LGR data. For $CH_4$, RMS error between the two instruments is 0.063 mmol m$^{-2}$ d$^{-1}$ using
numerically dry LGR data and 0.064 mmol m$^{-2}$ d$^{-1}$ using ambient LGR data.

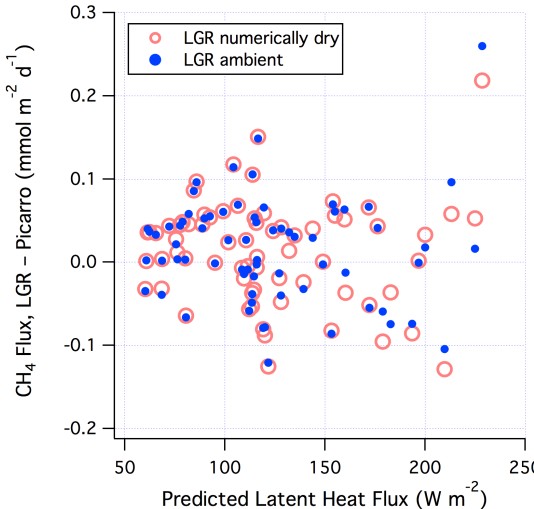

Figure 5. Difference in $CH_4$ flux between LGR (numerically dry as well as ambient) and Picarro (physically dry) vs the bulk
latent heat flux for the air-water wind sector of 45 to 80°. The numerical H$_2$O correction has the largest effect on $CH_4$ flux under
conditions of high latent heat flux.



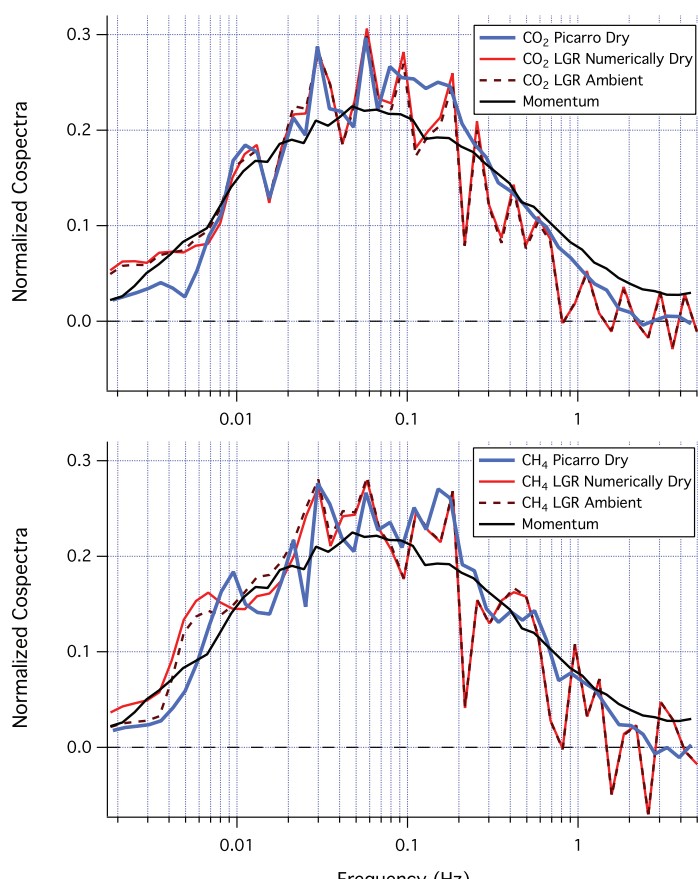

Figure 6. Mean cospectra of $CO_2$ (top), $CH_4$ (bottom) with momentum for the air-water wind sector of 45 to 80° during the Picarro (dry) vs LGR (wet) period, normalized by the respective mean flux. Gas cospectra from the two instruments are similar in the mean and show reasonable agreement with the momentum cospectrum. Spectral similarity suggests high frequency flux losses of ~6% for $CO_2$ and ~4% for $CH_4$ during this period.





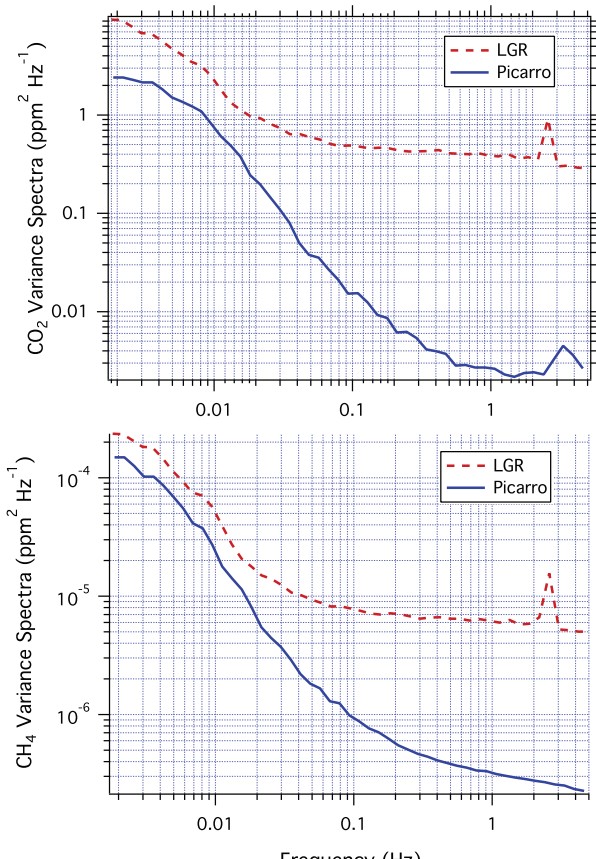

Figure 7. Mean variance spectra of dry $CO_2$ (top) and $CH_4$ (bottom) from the air-water wind sector during the Picarro (dry) vs LGR (wet) period. The LGR shows significantly greater variance, especially at high frequencies.




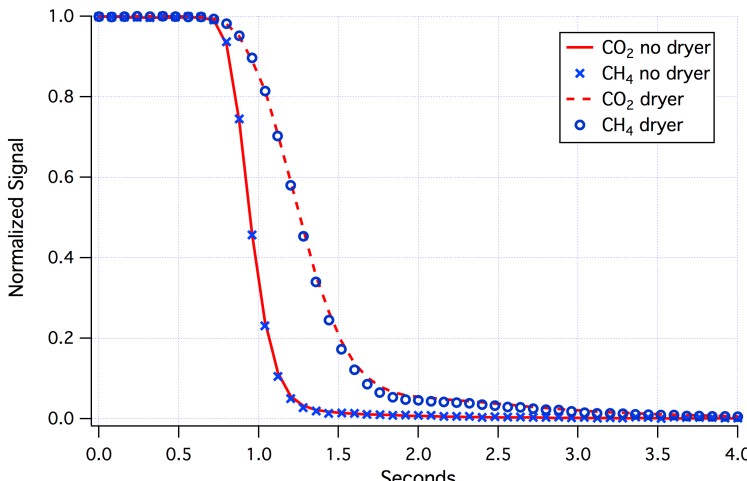

Figure 8. Fall offs in the Picarro dry $CO_2$ and $CH_4$ signals shortly after reductions in the calibration gas flow rate. The use of a Nafion dryer increases the observed response time by about 0.3 s.

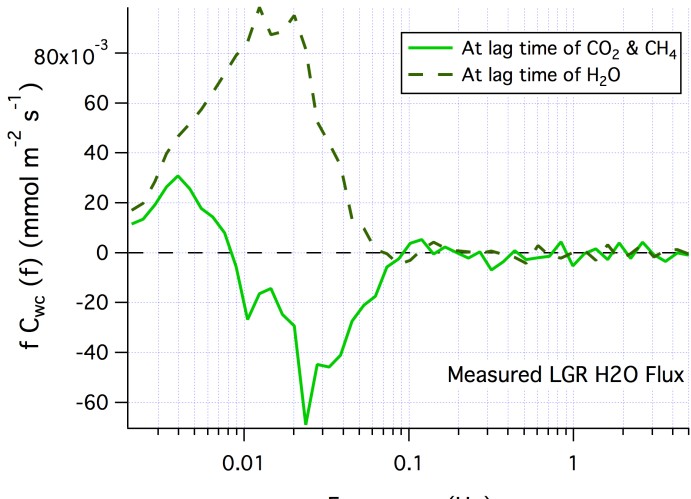

Figure 9. Mean cospectra of LGR $H_2O$ flux for the air-water wind sector, computed at the optimal lag time of $CO_2/CH_4$ (3 s) and the optimal lag time of $H_2O$ (~20 s). The much longer lag time of $H_2O$ is due to severe high frequency attenuation of $H_2O$ in the inlet tube (e.g. no flux signal above ~0.1 Hz). Even at the optimal lag time, measured $H_2O$ flux is only about 10% of the predicted latent heat flux (mean of 120 W m$^{-2}$).




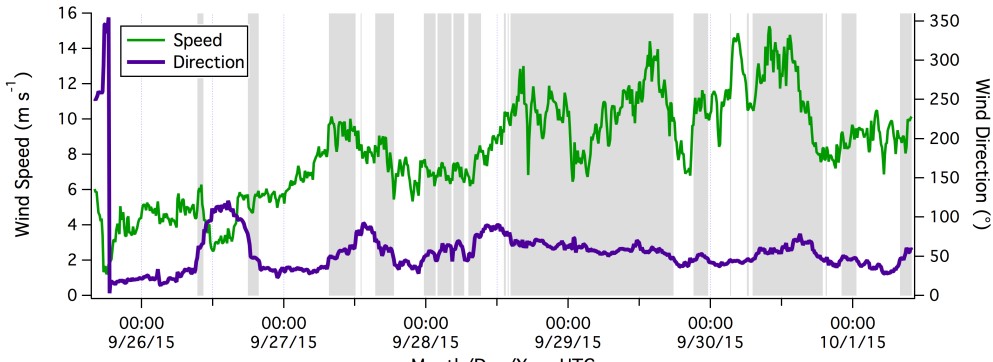

Figure A1. Time series of wind speed and direction during the Picarro (dry) vs LGR (wet) period. The shaded region indicates the wind sector between 45 and 80°, which we consider to be representative of air-water transfer.

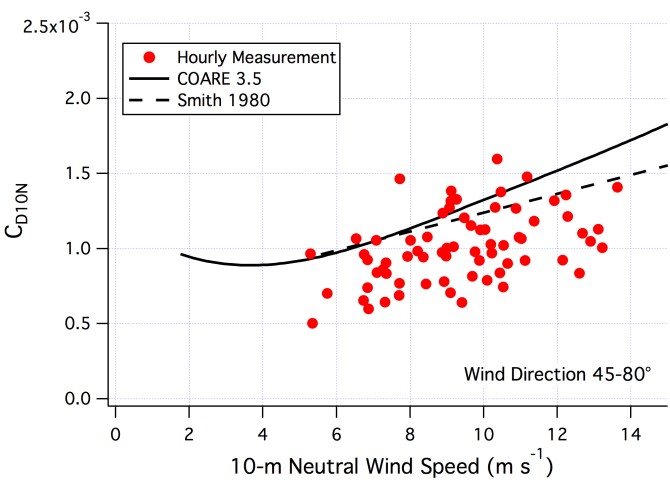

Figure A2. 10-m neutral drag coefficient vs. 10-m neutral wind speed for the air-water wind sector.

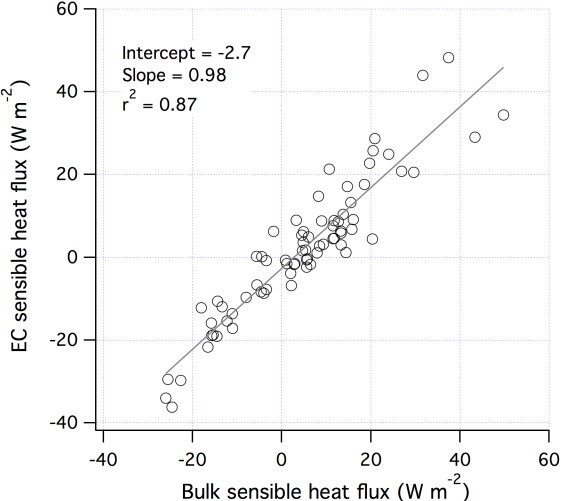

Figure A3. Measured vs. predicted sensible heat flux for the air-water wind sector.