# Peer review of "Comparison of two closed-path cavity based spectrometers for measuring air-water $CO_2$ and $CH_4$ fluxes by eddy covariance"

_Atmospheric Measurement Techniques, 2016_

## Referee Comment (RC1) · S. Landwehr (Referee) · 20 Sep 2016

**General comments**

In this submission the authors present direct Eddy Covariance measurements of $CO_2$ and $CH_4$ air-sea fluxes made with two commercially available cavity based spectrometers (a Picarro G2311-f and a Los Gatos Research (LGR) FGGA) on a coastal site. The two analysers are deployed side by side and the flux measurements are compared directly. The Picarro was most of the time deployed downstream of a Nafion dryer, while the LGR was sampling the air directly.

The authors address measurement issues like the under sampling of high frequency

fluctuations by the closed path Eddy Covariance systems and the cross-sensitivity of the optical measurements of $CO_2$ and $CH_4$ to $H_2O$.

Different methods are used to calculate flux detection limits for $CO_2$ and $CH_4$ and these are put in context with global climatologies, providing useful guidance for the planing of future measurements of these gas fluxes over the open ocean.

The authors find that the $CH_4$ and $CO_2$ fluxes measured by the two analysers agree within the given uncertainties, but that the LGR showed much higher noise in the two flux signals than the Picarro. A discussion of the potential reasons for the inferior performance of the LGR, like suboptimal cavity ringdown time, less rigorous maintenance of a stable measurement cell pressure and temperature, as well as a less sophisticated $H_2O$ cross sensitivity correction performed by the LGR is provided. In my opinion the addressing of the sample air density and $H_2O$ cross sensitivity related corrections made in the LGR would benefit from some minor revisions.

**Specific comments**

Lines 131–133 (page 5) "*As a result, we expect biases in $CO_2$ and $CH_4$ fluxes computed from Equations 1 and 2 only when the fluctuations in $H_2O$ are large and are correlated with fluctuations in dry $CO_2$ and $CH_4$ mixing ratios (i.e. due to any residual cross-sensitivity with $H_2O$).*":

I find this sentence rather confusing: In general the fluctuations of the concentrations of all three gases should be highly correlated as they are transported by the same eddies. Biases in the $CO_2$ and $CH_4$ fluxes computed from Equations 1 and 2 would suggest that the cross-sensitivity model is insufficient or that the coefficients are inaccurate, e.g., when $b = d = 0$ is assumed for the LGR FGGA.

The relative magnitudes of the corrections made in the Equations 1 and 2 scale with the magnitude of the $H_2O$ fluctuations (in the measurement volume) and with the ratio

of the $CO_2/CH_4$ background concentrations to the ambient fluxes, which is typically the case for $CO_2$.

Lines 155–159 (page 6): Based on the slow response to the flushing with pure nitrogen, I would speculated that the offset could be caused by $H_2O$ sticking to the mirrors of the LGR cavity (or rather to the salt and dust particles mentioned in lines 214–215). How did the Picarro react to the flushing with pure nitrogen?

Lines 196–198 (page 7) *"Since the gas fluxes were computed using the same wind data and the two analyzers were sampling the same gas stream, differences between them are primarily caused by noise in the instruments, rather than by the presence of water vapor."*:

Based on the evidence provided I cannot follow this conclusion. It might well be that the relatively small $H_2O$ cross sensitivity corrections, which are applied by the LGR, are insufficient. See also the next comment on Figure 5.

Lines 199–204 (page 7) and Figure 5.: If the differences in the $CH_4$ and $CO_2$ measurements from the LGR (wet) and Picarro (dry) are caused by cross sensitivity of the LGR signals to $H_2O$ one would expect a correlation with the latent heat flux measured by the LGR, but not necessarily with the predicted latent heat flux (the authors stated large and variable losses in the $H_2O$ flux signal measured by the LGR). I would therefore suggest to use the Latent heat flux measured by the LGR, instead of the predicted flux, as independent variable in Figure 5. The difference in the $CO_2$ fluxes measured by the two instruments should also be plotted as function of the latent heat flux measured by the LGR.

Lines 206–207 (page 7): How where the coefficients for the here mentioned spectral

line broadening correction for the LGR determined?

By using the Picarro (dry) $CH_4$ and $CO_2$ measurements as reference signal, you could calculate spectral line broadening coefficients for this specific LGR instrument in real time. Are these estimated coefficients constant or do they change in time? The latter might indicate a similar cross sensitivity effect as for the non-dispersive infra red gas analysers ( Prytherch et al. 2010, Blomquist et al. (2014), and Landwehr et al. (2014)).

Lines 214–215 (page 8): Did you get a chance to verify this by opening the cavity? The presence of salt and dust particles in the cavity might also explain the slow response to the flushing with N2, mentioned in the lines 155–159.

Lines 223–227 (page 8): Were the temperature and pressure in the cavities measured and used to account for dilution effects on $CH_4$ and $CO_2$ (Webb correction)?

Lines 301–305 (page 11): For the estimation of the high frequency loss in the gas fluxes, it might be more adequate to use the sensible/virtual heat flux cospectra measured by the (open-path) sonic anemometer, instead of the momentum flux cospectra.

Figure 3: Do the authors have any suggestions, what may cause the large scatter apparent in the difference between the LGR fluxes from numerically dry and ambient mixing ratios of $CO_2$ and $CH_4$, while the same difference appears to be a solemn function of the humidity flux for the Picarro?

Lines 354–357 (page 12) and Figure A2: I would suggest to add a trend line to the data shown in Figurer A2. To me the difference in the drag coefficients looks more like

30% of the CORARE 3.5 drag coefficient.

From comparing the Picarro (wet) and Picarro (dry) measurements can you find any effect of the Nafion dryer on the $CO_2$ and $CH_4$ flux detection limits?

**Technical corrections**

Lines 134–135 (page 5): I would suggest to add the uncertainties of the slopes and intercepts.

---

## Referee Comment (RC2) · B. Butterworth (Referee) · 21 Sep 2016

General Comments

In "Comparison of two closed-path cavity based spectrometers for measuring air-water CO2 and CH4 fluxes by eddy covariance" the authors compare the performance two commercially available gas analyzers (Picarro G2311-f and LGR FGGA) used to measuring CH4 and CO2 fluxes. As a research article on the intercomparison of gas measurement instruments this manuscript falls squarely into the scope of AMT. As stated in the article, instrument performance over terrestrial sites does not necessarily apply to ocean studies as the fluxes of CH4 and CO2 over the ocean are much smaller, justifying a study looking at air-sea fluxes. The findings will be of interest to anyone at-

tempting to make such field measurements, specifically for deciding which instrument is best suited to their needs. Paper provides a thorough description of the Picarro and LGR in action, highlighting certain technical characteristics of which future users should be aware (e.g., lack of line broadening correction applied internally on LGR, positive offset in dry $CO_2$ mixing ratio in LGR, etc.).

The findings are scientifically sound, following the necessary instrumental and processing techniques that have been developed over the past couple decades. The stationary platform avoids need to correct for motion. Intervals were filtered for non-stationarity. Estimates for high frequency flux loss were performed and magnitudes compare well with previous studies. Flux quality is verified by showing the mean cospectra for $CO_2$ and $CH_4$ fluxes against the momentum flux cospectra. Drying the airstream to at least one of the systems (as was done here) was important to rule out spurious $CO_2/CH_4$ flux due to water vapor. The paper was thorough, using different methods to estimate performance when possible (e.g., theoretical and empirical estimates for flux detection limits, filter function and ogive estimates for high-frequency loss, etc.).

With regard to the main thrust of the paper (comparing the performance of measuring flux from the two instruments) the results support the conclusions. The interpretations on the effectiveness of the numerical corrections for water vapor are not as strongly supported by the data. The paper should acknowledge the limitations of the dataset for specifically assessing the true impact of water vapor. An in-depth assessment of the effectiveness of the numerical corrections was not required for this paper, and could be the subject of another study. Some other minor issues/comments are discussed below.

Specific Comments

Line 101 – Inner diameter is more useful if anyone wanted to run the numbers (e.g., Reynolds number, expected flow rates, etc.).

Line 107 – It's unfortunate that the flushing time is slower than 10Hz. It would be good to know how much improvement the LGR might see at faster flow rate.

Line 127 – Not clear how you know that the LGR only applied dilution correction (and not line broadening) internally.

Line 177 – I don't understand how Figure 3 shows that "cross-sensitivities between $CO_2$ and $CH_4$ vs $H_2O$ fluxes are well accounted for by Equations 1 and 2." It seems to show only the impact of the corrections, not real information about how water vapor is influencing the flux measurements. The absence of "additional non-linearity" just means that the numerical corrections are generally linear (i.e., coefficients b and d are effectively zero). To determine the impact of water vapor would require having simultaneous flux measurements using identical instruments, with one dried and one undried. In Figure 2 it's clear that the difference between the LGR numerically corrected fluxes and LGR ambient fluxes is small compared to the difference between LGR and Picarro fluxes. Because the difference between the instruments is greater than the numerical correction for $H_2O$, it is difficult to assess the true impact of water vapor on the measurements. Presumably the impact of water vapor on the measured fluxes is not large due to the attenuation of the water vapor fluctuations in the long tube line (as stated in lines 147, 211) – the larger lag time in $H_2O$ should cause any real correlation between fluctuations in $H_2O$ and $CO_2$/$CH_4$ to become uncorrelated. But I would be wary of concluding that the numerical corrections are functioning properly when there is no actual way to verify with the dataset.

Line 198 – Same general comment as above – no clear proof that water vapor is not a factor.

Line 218 – Interesting that the scatter in hourly $CO_2$ flux from LGR was only 50% higher than Picarro, while Fig. 7 shows order of magnitude greater variance for LGR compared to Picarro. Any idea why this is? Is this just due to averaging?

Line 286 – While not large, tubing can cause some high frequency attenuation of $CO_2$ fluxes (Goulden et al. 1997, Ibrom et al. 2007). With an 18m tube it may not be insignificant. Of course, for the comparison of the two instruments it doesn't really

matter, since both will be measuring the same air.

Line 289 – It's not clear to me why reducing the flow rate will show response time. Reducing the flow of the calibration gas should not change the mixing ratio of the gas in the chamber. How does this work?

Line 305 – Does the Picarro (undried) show the same high frequency loss in the cospectra as Picarro (dried)? If so, the high frequency loss may be attributable to the long tube line, not just the nafion.

Based on the shorter lag time here than for the maximum covariance lag used in the CO2/CH4 calculations it appears that the time constants were found using a short bit of tube from the cal gas tank - through the nafion - then through the gas analyzer? Future measurement campaigns from this site would benefit from inlet testing (timed release of calibration gas in front of the inlet tube). Time constants for the whole system could be obtained, and used to estimate an overall high frequency flux loss. Also lag times could be measured directly, and be used to verify the maximum covariance lags.

Technical Corrections

Line 55 – is rigorously and quantitatively confirmed was rigorously and quantitatively confirmed

Line 108 – "LPM (at atmospheric pressure)" same as "SLPM", which was the notation used in the previous paragraph. Why not stay consistent?

Line 187 – Staionarity Stationarity

Line 218 – In this one sentence you report CH4 first, then CO2. Every other time it's CO2 followed by CH4 (even the next sentence). I would keep it consistent.

Line 234 – Switched from mmol m-2 d-1 to mmole m-2 d-1 for this section. Keep consistent.

Line 273 – (Kort et al. (2012) (Kort et al. 2012)

Line 351 – Unnecessary ")" at end of CD10N equation.

Line 358 – Water vapor correction based on Schotanus et al. (1983)?

References

Schotanus, P., F. T. M. Nieuwstadt, and H. A. R. de Bruin, 1983: Temperature Measurement with a Sonic Anemometer and its Application to Heat and Moisture Fluxes. Boundary- Layer Meteorol., 26, 81–93.

Goulden, M.L., B.C. Daube, S.-M. Fan, D.J. Sutton, A. Bazzaz, J.W. Munger, S.C. Wofsy, 1997: Physiological responses of a Black Spruce forest to weather. J. Geophys. Res., 102, 28,987–128,996.

Ibrom, A., E. Dellwik, H. Flyvbjerg, N. O. Jensen, and K. Pilegaard, 2007: Strong low-pass filtering effects on water vapour flux measurements with closed-path eddy correlation systems. Agric. For. Meteorol., 147, 140–156.

---

## Referee Comment (RC3) · S. Landwehr (Referee) · 21 Sep 2016

**Corrigendum**

I wrote my review based on the manuscript uploaded to AMT 27 Jun 2016, which has continuous line numbering and was provided to the reviewers. The publicly available discussion version amt-2016-215.pdf is identical to this original manuscript, however the line numbering is reset for every page. Below is the same review but with adapted line numbering to match the discussion version.

**General comments**

In this submission the authors present direct Eddy Covariance measurements of $CO_2$ and $CH_4$ air-sea fluxes made with two commercially available cavity based spectrometers (a Picarro G2311-f and a Los Gatos Research (LGR) FGGA) on a coastal site. The two analysers are deployed side by side and the flux measurements are compared directly. The Picarro was most of the time deployed downstream of a Nafion dryer, while the LGR was sampling the air directly.

The authors address measurement issues like the under sampling of high frequency fluctuations by the closed path Eddy Covariance systems and the cross-sensitivity of the optical measurements of $CO_2$ and $CH_4$ to $H_2O$.

Different methods are used to calculate flux detection limits for $CO_2$ and $CH_4$ and these are put in context with global climatologies, providing useful guidance for the planing of future measurements of these gas fluxes over the open ocean.

The authors find that the $CH_4$ and $CO_2$ fluxes measured by the two analysers agree within the given uncertainties, but that the LGR showed much higher noise in the two flux signals than the Picarro. A discussion of the potential reasons for the inferior performance of the LGR, like suboptimal cavity ringdown time, less rigorous maintenance of a stable measurement cell pressure and temperature, as well as a less sophisticated $H_2O$ cross sensitivity correction performed by the LGR is provided. In my opinion the addressing of the sample air density and $H_2O$ cross sensitivity related corrections made in the LGR would benefit from some minor revisions.

**Specific comments**

Page 5, lines 8–10 "*As a result, we expect biases in $CO_2$ and $CH_4$ fluxes computed from Equations 1 and 2 only when the fluctuations in $H_2O$ are large and are correlated with fluctuations in dry $CO_2$ and $CH_4$ mixing ratios (i.e. due to any residual cross-sensitivity with $H_2O$).*":

I find this sentence rather confusing: In general the fluctuations of the concentrations of all three gases should be highly correlated as they are transported by the same eddies. Biases in the $CO_2$ and $CH_4$ fluxes computed from Equations 1 and 2 would suggest that the cross-sensitivity model is insufficient or that the coefficients are inaccurate, e.g., when $b = d = 0$ is assumed for the LGR FGGA.

The relative magnitudes of the corrections made in the Equations 1 and 2 scale with the magnitude of the $H_2O$ fluctuations (in the measurement volume) and with the ratio of the $CO_2$/$CH_4$ background concentrations to the ambient fluxes, which is typically the case for $CO_2$.

Page 6, lines 3–6: Based on the slow response to the flushing with pure nitrogen, I would speculated that the offset could be caused by $H_2O$ sticking to the mirrors of the LGR cavity (or rather to the salt and dust particles mentioned in lines 214–215). How did the Picarro react to the flushing with pure nitrogen?

Page 7, lines 15–17 *"Since the gas fluxes were computed using the same wind data and the two analyzers were sampling the same gas stream, differences between them are primarily caused by noise in the instruments, rather than by the presence of water vapor."*:

Based on the evidence provided I cannot follow this conclusion. It might well be that the relatively small $H_2O$ cross sensitivity corrections, which are applied by the LGR, are insufficient. See also the next comment on Figure 5.

Page 7, lines 18–23 and Figure 5.: If the differences in the $CH_4$ and $CO_2$ measurements from the LGR (wet) and Picarro (dry) are caused by cross sensitivity of the LGR signals to $H_2O$ one would expect a correlation with the latent heat flux measured by the LGR, but not necessarily with the predicted latent heat flux (the authors stated large

and variable losses in the $H_2O$ flux signal measured by the LGR). I would therefore suggest to use the Latent heat flux measured by the LGR, instead of the predicted flux, as independent variable in Figure 5. The difference in the $CO_2$ fluxes measured by the two instruments should also be plotted as function of the latent heat flux measured by the LGR.

Page 7, lines 25–26: How where the coefficients for the here mentioned spectral line broadening correction for the LGR determined?

By using the Picarro (dry) $CH_4$ and $CO_2$ measurements as reference signal, you could calculate spectral line broadening coefficients for this specific LGR instrument in real time. Are these estimated coefficients constant or do they change in time? The latter might indicate a similar cross sensitivity effect as for the non-dispersive infra red gas analysers ( Prytherch et al. 2010, Blomquist et al. (2014), and Landwehr et al. (2014)).

Page 8, lines 3–4: Did you get a chance to verify this by opening the cavity? The presence of salt and dust particles in the cavity might also explain the slow response to the flushing with N2, mentioned in the lines 155–159.

Page 8, lines 12–16: Were the temperature and pressure in the cavities measured and used to account for dilution effects on $CH_4$ and $CO_2$ (Webb correction)?

Page 11, lines 1–5: For the estimation of the high frequency loss in the gas fluxes, it might be more adequate to use the sensible/virtual heat flux cospectra measured by the (open-path) sonic anemometer, instead of the momentum flux cospectra.

Figure 3: Do the authors have any suggestions, what may cause the large scatter

apparent in the difference between the LGR fluxes from numerically dry and ambient mixing ratios of $CO_2$ and $CH_4$, while the same difference appears to be a solemn function of the humidity flux for the Picarro?

Page 12, lines 24–27 and Figure A2: I would suggest to add a trend line to the data shown in Figurer A2. To me the difference in the drag coefficients looks more like 30% of the CORARE 3.5 drag coefficient.

From comparing the Picarro (wet) and Picarro (dry) measurements can you find any effect of the Nafion dryer on the $CO_2$ and $CH_4$ flux detection limits?

**Technical corrections**

Page 5, lines 11-12:: I would suggest to add the uncertainties of the slopes and intercepts.

---

## Author Comment (AC1) · 26 Oct 2016

**Comparison of two closed-path cavity based spectrometers for measuring air-water CO2 and CH4 fluxes by eddy covariance**

by M. Yang et al.

26 October, 2016

Many thanks for the detailed and insightful *comments and suggestions from Referee S. Landwehr*.  Below we present each comment (in *italic*), followed by our reply.  All of our replies are incorporated into the revised manuscript where appropriate, unless indicated otherwise.

**General comments**
*In this submission the authors present direct Eddy Covariance measurements of CO2 and CH4 air-sea fluxes made with two commercially available cavity based spectrometers (a Picarro G2311-f and a Los Gatos Research (LGR) FGGA) on a coastal site. The two analysers are deployed side by side and the flux measurements are compared directly. The Picarro was most of the time deployed downstream of a Nafion dryer, while the LGR was sampling the air directly.*
*The authors address measurement issues like the under sampling of high frequency fluctuations by the closed path Eddy Covariance systems and the cross-sensitivity of the optical measurements of CO2 and CH4 to H2O.*
*Different methods are used to calculate flux detection limits for CO2 and CH4 and these are put in context with global climatologies, providing useful guidance for the planning of future measurements of these gas fluxes over the open ocean.*
*The authors find that the CH4 and CO2 fluxes measured by the two analysers agree within the given uncertainties, but that the LGR showed much higher noise in the two flux signals than the Picarro. A discussion of the potential reasons for the inferior performance of the LGR, like suboptimal cavity ringdown time, less rigorous maintenance of a stable measurement cell pressure and temperature, as well as a less sophisticated H2O cross sensitivity correction performed by the LGR is provided. In my opinion the addressing of the sample air density and H2O cross sensitivity related corrections made in the LGR would benefit from some minor revisions.*

We are happy to hear that the referee finds out contribution useful.  Our answers below and changes in the manuscript address the referee's specific questions.

**Specific comments**
*Page 5, lines 8–10 "As a result, we expect biases in CO2 and CH4 fluxes computed from Equations 1 and 2 only when the fluctuations in H2O are large and are correlated with fluctuations in dry CO2 and CH4 mixing ratios (i.e. due to any residual cross-sensitivity with H2O).":*
*I find this sentence rather confusing: In general the fluctuations of the concentrations of all three gases should be highly correlated as they are transported by the same eddies.*

*Biases in the CO2 and CH4 fluxes computed from Equations 1 and 2 would suggest that the cross-sensitivity model is insufficient or that the coefficients are inaccurate, e.g., when b = d = 0 is assumed for the LGR FGGA.*
*The relative magnitudes of the corrections made in the Equations 1 and 2 scale with the magnitude of the H2O fluctuations (in the measurement volume) and with the ratio of the CO2/CH4 background concentrations to the ambient fluxes, which is typically the case for CO2.*

We agree with the referee's comment and apologize for our confusing statement. We have now modified that sentence to "Significant biases in CO2 and CH4 fluxes computed from Equations 1 and 2 would imply that these mathematical corrections are inaccurate or insufficient at describing the cross-sensitivities between the trace gases and H2O. The greatest relative biases are expected to occur when the magnitude of the H2O fluctuations in the measurement cavity is large and when the trace gas fluxes are small."

*Page 6, lines 3–6: Based on the slow response to the flushing with pure nitrogen, I would speculated that the offset could be caused by H2O sticking to the mirrors of the LGR cavity (or rather to the salt and dust particles mentioned in lines 214–215). How did the Picarro react to the flushing with pure nitrogen?*

Thanks for the comment. The Picarro CO2, CH4, and H2O levels approached zero very rapidly when measuring pure nitrogen. The LGR does exhibit a positive offset relative to the Picarro in the H2O measurement as well (by ~100 ppm when measuring N2), but this difference is not nearly large enough to cause a bias in CO2 measurement on the order of ~10 ppm. More recent measurements of NOAA CO2 gas standards using the LGR show that the offset in the LGR is most likely due to an inaccurate instrument calibration.

*Page 7, lines 15–17 "Since the gas fluxes were computed using the same wind data and the two analyzers were sampling the same gas stream, differences between them are primarily caused by noise in the instruments, rather than by the presence of water vapor." :*
*Based on the evidence provided I cannot follow this conclusion. It might well be that the relatively small H2O cross sensitivity corrections, which are applied by the LGR, are insufficient. See also the next comment on Figure 5.*

We made this statement because the LGR fluxes (wet) and the Picarro fluxes (physically dried) are similar in the mean. The LGR fluxes (wet and numerically dried) show greater scatter, which is primarily due to noise in the LGR instrument rather than any corrections for water vapor. To be more specific, we have changed the second part of the sentence to "…hour to hour differences between them are primarily…."

*Page 7, lines 18–23 and Figure 5.: If the differences in the CH4 and CO2 measurements from the LGR (wet) and Picarro (dry) are caused by cross sensitivity of the LGR signals to H2O one would expect a correlation with the latent heat flux measured by the LGR, but not necessarily with the predicted latent heat flux (the authors stated large and variable losses in the H2O flux signal measured by the LGR). I would therefore suggest*

*to use the Latent heat flux measured by the LGR, instead of the predicted flux, as independent variable in Figure 5. The difference in the CO2 fluxes measured by the two instruments should also be plotted as function of the latent heat flux measured by the LGR.*

Thanks a lot for the comment. The plots below show what the referee had suggested. The LGR H2O flux is computed at the optimal lag time for H2O (~20s). The same qualitative trend is seen as in Fig. 5, i.e. the differences in CO2 and CH4 fluxes due to the H2O correction increase with increasing H2O flux. Also, the equivalent of the current Fig. 5 for CO2 illustrates a similar pattern to CH4, which is why we didn't show it in the current manuscript.

[Figure]

We did not include these plots in the paper because the measured LGR H2O flux was severely attenuated by the tubing and its actual magnitude in W/m2 is probably not especially informative for other users/translatable to other setup. The following plot shows that measured LGR H2O flux increases non-linearly with bulk H2O flux. We will add these plots to supplementary materials.

[Figure]

*Page 7, lines 25–26: How where the coefficients for the here mentioned spectral line broadening correction for the LGR determined?*

We used coefficients proposed by Hiller et al. 2012 here. This is now specified in the paper.

*By using the Picarro (dry) CH4 and CO2 measurements as reference signal, you could calculate spectral line broadening coefficients for this specific LGR instrument in real time. Are these estimated coefficients constant or do they change in time? The latter might indicate a similar cross sensitivity effect as for the non-dispersive infra red gas analysers ( Prytherch et al. 2010, Blomquist et al. (2014), and Landwehr et al. (2014)).*

It'd likely be rather uncertain to use the Picarro (dry) data to determine the spectral broadening coefficients for the LGR due to the existing instrumental calibration offsets between the Picarro and LGR at measuring dry CO2/CH4.
To more accurately determine the spectral broadening coefficients, one could look at the LGR responses while humidifying a CO2/CH4 gas standard to different humidity levels (monitored by a separate humidity sensor). Such a calibration would be more beneficial to high-precision CO2/CH4 mixing ratio measurements than to this work, considering the likely very small effect of this correction on the CO2/CH4 fluxes.

*Page 8, lines 3–4: Did you get a chance to verify this by opening the cavity? The presence of salt and dust particles in the cavity might also explain the slow response to the flushing with N2, mentioned in the lines 155–159.*

We did open the cavity. While dust and sea salt were not clearly visible to the naked eye, the simple act of opening the cavity and closing it again further reduced the ringdown time, suggesting that the mirrors became more contaminated by exposure to the hut air.

*Page 8, lines 12–16: Were the temperature and pressure in the cavities measured and used to account for dilution effects on CH4 and CO2 (Webb correction)?*

Temperature and pressure of the cavities were continuously monitored (at 10 Hz). The CO2 and CH4 fluxes were computed from the measured mixing ratios at these T, P (rather than from mass concentrations), so that an additional Webb correction for T, P shouldn't be necessary. The differences between LGR wet and numerically dried fluxes do include the dilution effect due to humidity.

*Page 11, lines 1–5: For the estimation of the high frequency loss in the gas fluxes, it might be more adequate to use the sensible/virtual heat flux cospectra measured by the (open-path) sonic anemometer, instead of the momentum flux cospectra.*

We agree that heat flux, being a scalar flux, is more commonly used for the quantification of the high frequency flux loss. During this period the sensible heat flux was fairly small and fluctuated in sign between day and night. As a result the mean sensible heat flux spectrum was very noisy, which is why we chose to compare to momentum cospectrum instead.

*Figure 3: Do the authors have any suggestions, what may cause the large scatter*

*apparent in the difference between the LGR fluxes from numerically dry and ambient mixing ratios of CO2 and CH4, while the same difference appears to be a solemn function of the humidity flux for the Picarro?*

This is likely due to the greater noise in the LGR instrument.

*Page 12, lines 24–27 and Figure A2: I would suggest to add a trend line to the data shown in Figurer A2. To me the difference in the drag coefficients looks more like 30% of the CORARE 3.5 drag coefficient.*

Suggestion accepted.  Please see below.  The apparent underestimation of the measurement is ~15% at low wind speeds and ~30% at the highest wind speed.  This is probably due to flow distortion by the local topography (i.e. mostly speeding up of horizontal wind), which we have not accounted for.

[Figure]

*From comparing the Picarro (wet) and Picarro (dry) measurements can you find any effect of the Nafion dryer on the CO2 and CH4 flux detection limits?*

The high frequency noise levels in CO2 and CH4 in the Picarro are comparable with and without the Nafion dryer.  Thus we do not expect a significant difference in the Picarro detection limits due to the presence/absence of the dryer.

**Technical corrections**
*Page 5, lines 11-12:: I would suggest to add the uncertainties of the slopes and intercepts.*
The uncertainties of the slopes (1 standard deviation) were about 0.001 (i.e. 0.1% since the slopes were close to unity).  The uncertainties of the intercepts were about 0.5 ppm for CO2 and 0.0018 ppm for CH4 (i.e. also ~0.1% of the respective ambient mixing ratios).

---

## Author Comment (AC2) · 26 Oct 2016

Please see attachment.

Please also note the supplement to this comment:
http://www.atmos-meas-tech-discuss.net/amt-2016-215/amt-2016-215-AC2-supplement.pdf